# UCSG-NET - Unsupervised Discovering of Constructive Solid Geometry Tree

**Kacper Kania**[1]*     **Maciej Zięba**[1,2]     **Tomasz Kajdanowicz**[1]

[1]Wrocław University of Science and Technology
[2]Tooploox
kacp.kania@gmail.com

## Abstract

Signed distance field (SDF) is a prominent implicit representation of 3D meshes. Methods that are based on such representation achieved state-of-the-art 3D shape reconstruction quality. However, these methods struggle to reconstruct non-convex shapes. One remedy is to incorporate a constructive solid geometry framework (CSG) that represents a shape as a decomposition into primitives. It allows to embody a 3D shape of high complexity and non-convexity with a simple tree representation of Boolean operations. Nevertheless, existing approaches are supervised and require the entire CSG parse tree that is given upfront during the training process. On the contrary, we propose a model that extracts a CSG parse tree without any supervision - UCSG-NET. Our model predicts parameters of primitives and binarizes their SDF representation through differentiable indicator function. It is achieved jointly with discovering the structure of a Boolean operators tree. The model selects dynamically which operator combination over primitives leads to the reconstruction of high fidelity. We evaluate our method on 2D and 3D autoencoding tasks. We show that the predicted parse tree representation is interpretable and can be used in CAD software.[1]

## 1 Introduction

Neural networks for 3D shape analysis gained much popularity in recent years. Among their main advantages are fast inference for unknown shapes and high generalization power. Many approaches rely on the different representations of the input: implicit such as voxel grids, point clouds and signed distance fields [1–3], or explicit - meshes [4]. Meshes can be found in computer-aided design applications, where a graphic designer often composes complex shapes out simple shapes primitives, such as boxes and spheres.

Existing methods for representing meshes, such as BSP-NET [5] and CVXNET [6], achieve remarkable accuracy on a reconstruction tasks. However, the process of generating the mesh from predicted planes requires an additional post-processing step. These methods also assume that any object can be decomposed into a union of convex primitives. While holding, it requires many such primitives to represent concave shapes. Consequently, the decoding process is difficult to explain and modified with some external expert knowledge. On the other hand, there are fully interpretable approaches, like CSG-NET [7, 8], that utilize CSG parse tree to represent 3D shape construction process. Such solutions require expensive supervision that assumes assigned CSG parse tree for each example given during training.

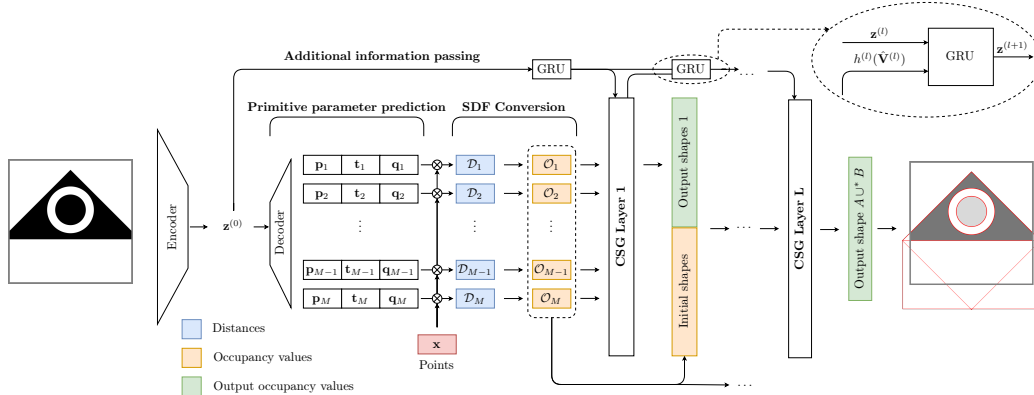

Figure 1: Pipeline of the UCSG-NET. Example of CSG layer is shown in Figure 3.

In this work, we propose a novel model for representing 3D meshes capable of learning CSG parse trees in an unsupervised manner - UCSG-NET. We achieve the stated goal by introducing so-called CSG Layers capable of learning explainable Boolean operations for pairs primitives. CSG Layers create the interpretable network of the geometric operations that produce complex shapes from a limited number of simple primitives. We evaluate the representation capabilities of meshes of our approach using challenging 2D and 3D datasets. We summarize our main contributions as:

- Our method is the first one that is able to predict CSG tree without any supervision and achieve state-of-the-art results on the 2D reconstruction task comparing to CSG-NET trained in a supervised manner. Predictions of our method are fully interpretable and can aid in CAD applications.

- We define and describe a novel formulation of constructive solid geometry operations for occupancy value representation for 2D and 3D data.

## 2 Method

We propose an end-to-end neural network model that predicts parameters of simple geometric primitives and their constructive solid geometry composition to reconstruct a given object. Using our approach, one can predict the CSG parse tree that can be further passed to an external rendering software in order to reconstruct the shape. To achieve this, our model predicts primitive shapes in SDF representation. Then, it converts them into occupancy values $\mathcal{O}$ taking 1 if a point in the 2D or the 3D space is inside the shape and 0 otherwise. CSG operations on such a representation are defined as clipped summations and differences of binary values. The model dynamically chooses which operation should be used. During the validation, we retrieve the predicted CSG parse tree and shape primitives, and pass them to the rendering software. Thus, we need a single point in 3D space to infer the structure of the CSG tree. It is possible since primitive parameters and CSG operations are predicted independently from sampled points. In the following subsections, we present 2D examples for clarity. The method scales to 3D inputs trivially.

### 2.1 Constructive Solid Geometry Network

The UCSG-NET architecture is provided in Figure 1. The model is composed of the following main components: encoder, primitive parameter prediction network, signed distance field to indicator function converter and constructive solid geometry layers.

**Encoder** We process the input object $\mathcal{I}$ by mapping it into low dimensional latent vector $\mathbf{z}$ of length $d_{\mathbf{z}}$ using an encoder $f_\theta$, e.g. $f_\theta(\mathcal{I}) = \mathbf{z}$. Depending on the data type, we use either a 2D or 3D convolutional neural network as an encoder. The latent vector is then passed to the primitive parameter prediction network.

**Primitive parameter prediction network** The role of this component is to extract the parameters of the primitives, given the latent representation of the input object. The primitive parameter prediction network $g_\phi$ consists of multiple fully connected layers interleaved with activation functions. The last layer predicts parameters of primitives in the SDF representation. We consider primitives such as boxes and spheres that allow us to calculate signed distance analytically. We note that planes can be used as well, thus extending approaches like BSP-NET [5] and CVXNET [6]. The mathematical formulation of used shapes is provided in the supplementary material. The network produces $N$ tuples of $\{i \in N | \mathbf{p}_i, \mathbf{t}_i, \mathbf{q}_i\}$. $\mathbf{p}_i \in \mathbb{R}^{d_\mathbf{p}}$ describes vector of parameters of a particular shape (ex. radius of a sphere), while $\mathbf{t}_i \in \mathbb{R}^{d_\mathbf{t}}$ is the translation of the shape and $\mathbf{q}_i \in \mathbb{R}^{d_\mathbf{q}}$ - the rotation represented as a quaternion for 3D shapes and a matrix for 2D shapes. We further combine $k$ different shapes to be predicted by using a fully connected layer for each shape type separately, thus producing $kN = M$ shapes and $M \times (d_\mathbf{p} + d_\mathbf{t} + d_\mathbf{q})$ parameters in total.

Once parameters are predicted, we use them to calculate signed distance values for sampled points $\mathbf{x}$ from volume of space that boundaries are normalized to unit square (or unit cube for 3D data). For each shape, that has an analytical equation $dist$ parametrized by $\mathbf{p}$ that calculates signed distance from a point $\mathbf{x}$ to its surface, we obtain $\mathcal{D}_i = dist(\mathbf{q}_i^{-1}(\mathbf{x} - \mathbf{t}_i); \mathbf{p}_i)$.

**Signed Distance Field to Indicator Function Converter** CSG operations in SDF representation are often defined as a combination of min and max functions on distance values. One has to apply either LogSumExp operation as in CVXNET or standard Softmax function to obtain differentiable approximation. However, we cast our problem to predict CSG operations for occupancy-valued sets. The motivation is that these are linear operations, hence they provide better training stability.

We transform signed distances $\mathcal{D}$ to occupancy values $\mathcal{O} \in \{0, 1\}$. We use parametrized $\alpha$ clipping function that is learned with the rest of the pipeline:

$$\mathcal{O} = \left[ 1 - \frac{\mathcal{D}}{\alpha} \right]_{[0,1]} \begin{cases} \texttt{inside}, & \mathcal{O} = 1 \\ \texttt{outside}, & \mathcal{O} \in [0, 1) \end{cases} \tag{1}$$

where $\alpha$ is a learnable scalar and $\alpha > 0$, $[\cdot]_{[0,1]}$ clips values to the given range and $\mathcal{O}$ means an approximation of occupancy values. $\mathcal{O} = 1$ indicates the inside and the surface of a shape. $\mathcal{O} \in [0, 1)$ means outside of the shape and $\lim_{\alpha \to 0} \mathcal{O} \in \{0, 1\}$. Gradual learning of $\alpha$ allows to distribute gradients to all shapes in early stages of training. There are no specific restrictions for $\alpha$ initialization and we set $\alpha = 1$ in our experiments. The value is pushed towards 0 by optimizing jointly with the rest of parameters by adding the $|\alpha|$ term to the optimized loss. The method follows findings of Sakr et al. [9] that increasing slope of clipping function can be used to obtain binary activations.

**Constructive Solid Geometry Layer** Predicted sets of occupancy values and output of the encoder $\mathbf{z}$ are passed to a sequence of $L \geq 1$ CSG layers that combine shapes using boolean operators: union (denoted by $\cup^*$), intersection ($\cap^*$) and difference ($-^*$). To grasp an idea of how CSG is performed on occupancy-valued sets, we show example operations in Figure 2. CSG operations for two sets $A$ and $B$ are described as:

$$\begin{aligned} A \cup^* B = [A + B]_{[0,1]} && A -^* B = [A - B]_{[0,1]} \\ A \cap^* B = [A + B - 1]_{[0,1]} && B -^* A = [B - A]_{[0,1]} \end{aligned} \tag{2}$$

The question is how to choose operands $A$ and $B$, denoted as $\texttt{left}$ and $\texttt{right}$ operands, from input shapes $O^{(l)}$ that would compose the output shape in $O^{(l+1)}$. We create two learnable matrices $\mathbf{K}_{\texttt{left}}^{(l)}, \mathbf{K}_{\texttt{right}}^{(l)} \in \mathbb{R}^{M \times d_\mathbf{z}}$. Vectors stored in rows of these matrices serve as keys for a query $\mathbf{z}$ to select appropriate shapes for all 4 operations. The input latent code $\mathbf{z}$ is used as a query to retrieve the most appropriate operand shapes for each layer. We perform dot product between matrices $\mathbf{K}_{\texttt{left}}^{(l)}, \mathbf{K}_{\texttt{right}}^{(l)}$ and $\mathbf{z}$, and compute softmax along $M$ input shapes.

$$\mathbf{V}_{\texttt{left}}^{(l)} = \text{softmax}(\mathbf{K}_{\texttt{left}}^{(l)} \mathbf{z}) \qquad \mathbf{V}_{\texttt{right}}^{(l)} = \text{softmax}(\mathbf{K}_{\texttt{right}}^{(l)} \mathbf{z}) \tag{3}$$

The index of a particular operand is retrieved using Gumbel-Softmax [10] reparametrization of the categorical distribution:

$$\hat{V}_{\texttt{side},i}^{(l)} = \frac{\exp\left( \left( \log(V_{\texttt{side},i}^{(l)}) + c_i \right) / \tau^{(l)} \right)}{\sum_{j=1}^{M} \exp\left( \left( \log(V_{\texttt{side},j}^{(l)}) + c_j \right) / \tau^{(l)} \right)} \qquad \begin{aligned} &\text{for } i = 1, ..., M \\ &\text{and } \texttt{side} \in \{\texttt{left}, \texttt{right}\} \end{aligned} \tag{4}$$

$A \cup^* B :$ $\left[ \Box + \Box \right]_{[0,1]} = \Box$

$A \cap^* B :$ $\left[ \Box + \Box - \mathbf{1} \right]_{[0,1]} = \Box$

$A -^* B :$ $\left[ \Box - \Box \right]_{[0,1]} = \Box$

$B -^* A :$ $\left[ \Box - \Box \right]_{[0,1]} = \Box$

Figure 2: Example of constructive solid geometry on occupancy-valued sets

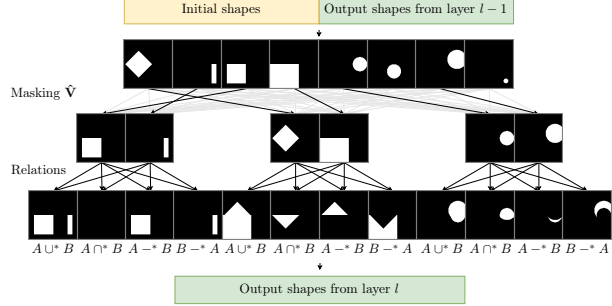

Figure 3: Example of relation layer prediction. The CSG layer chooses pairs operands by masking input shapes and performs all Boolean operations on selected shapes.

where $c_i$ is a sample from Gumbel(0, 1). The benefit of the reparametrization is twofold. Firstly, the expectation over the distribution stays the same despite changing $\tau^{(l)}$. Secondly, we can manipulate $\tau^{(l)}$ so for $\tau^{(l)} \to 0$ the distribution degenerates to categorical distribution. Hence, a single shape selection replaces the fuzzy sum of all input shapes in that case. That way, we allow the network to select the most appropriate shape for the composition during learning by decreasing $\tau^{(l)}$ gradually. By the end of the learning process, we can retrieve a single shape to be used for the CSG. The temperature $\tau^{(l)}$ is learned jointly with the rest of the parameters. Left and right operands $\mathcal{O}_{\texttt{left}}^{(l)}, \mathcal{O}_{\texttt{right}}^{(l)}$ are retrieved as:

$$\mathcal{O}_{\texttt{right}}^{(l)} = \sum_{i=1}^{M} \mathcal{O}_i^{(l)} \hat{V}_{\texttt{right},i}^{(l)} \qquad \mathcal{O}_{\texttt{left}}^{(l)} = \sum_{i=1}^{M} \mathcal{O}_i^{(l)} \hat{V}_{\texttt{left},i}^{(l)} \tag{5}$$

A set of output shapes from the $l+1$ CSG layer is obtained by performing all operations in Equation 2 on selected operands:

$$\mathcal{O}_{A \cup^* B}^{(l+1)} = \left[ \mathcal{O}_{\texttt{left}}^{(l)} + \mathcal{O}_{\texttt{right}}^{(l)} \right]_{[0,1]} \qquad\qquad \mathcal{O}_{A -^* B}^{(l+1)} = \left[ \mathcal{O}_{\texttt{left}}^{(l)} - \mathcal{O}_{\texttt{right}}^{(l)} \right]_{[0,1]}$$
$$\mathcal{O}_{A \cap^* B}^{(l+1)} = \left[ \mathcal{O}_{\texttt{left}}^{(l)} + \mathcal{O}_{\texttt{right}}^{(l)} - 1 \right]_{[0,1]} \qquad \mathcal{O}_{B -^* A}^{(l+1)} = \left[ \mathcal{O}_{\texttt{right}}^{(l)} - \mathcal{O}_{\texttt{left}}^{(l)} \right]_{[0,1]} \tag{6}$$

$$\mathcal{O}^{(l+1)} = \left[ \mathcal{O}_{A \cup^* B}^{(l+1)}; \mathcal{O}_{A \cap^* B}^{(l+1)}; \mathcal{O}_{A -^* B}^{(l+1)}; \mathcal{O}_{B -^* A}^{(l+1)} \right] \tag{7}$$

where $\texttt{left}, \texttt{right} \in M$ denotes left and right operands of the operation. By performing these operations manually, we increase the diversity of possible shape combinations and leave to the model which operations should be used for the reconstruction. Operations can be repeated to output multiple shapes. Note that the computation overhead increases linearly with the number of output shapes per layer. The whole procedure can be stacked in $l \leq L$ layers to create a CSG network. The $L$-th layer outputs a union since it is guaranteed to return a non-empty shape in most cases.

At this point, the network has to learn passing primitives untouched by operators if any primitive should be used in later layers of the CSG tree to create, for example, nested rings. To mitigate the problem, each $l+1$ layer receives outputs from the $l$-th layer concatenated with the original binarized values $\mathcal{O}^{(0)}$. For the first layer $l = 1$, it means receiving initial shapes only.

**Additional information passing** The information about what is left to reconstruct changes layer by layer. Therefore, we incorporate it into the latent code to improve the reconstruction quality and stabilize training. Firstly, we encode $\hat{\mathbf{V}}^{(l)} = [\hat{\mathbf{V}}_{\texttt{left}}^{(l)}; \hat{\mathbf{V}}_{\texttt{right}}^{(l)}]$ with a neural network $h^{(l)}$ containing a single hidden layer. Then, we employ GRU unit [11] that takes the latent code $\mathbf{z}^{(l)}$ and encoded $\hat{\mathbf{V}}^{(l)}$ as an input, and outputs the updated latent code $\mathbf{z}^{(l+1)}$ for the next layer. The hidden state of the GRU unit is learnable. The initial $\mathbf{z}^{(0)}$ is the output from the encoder.

**Interpretability** All introduced components of the UCSG-NET lead us to interpretable predictions of mesh reconstructions. To see this, consider the following case. When $\alpha \approx 0$, we obtain occupancy values calculated with Equation 1. Thus, shapes represented as these values will occupy the same

volume as meshes reconstructed from parameters $\{i \in M | \mathbf{p}_i, \mathbf{t}_i, \mathbf{q}_i\}$. These meshes can be visualized and edited explicitly. To further combine these primitives through CSG operations, we calculate $\arg\max_{i \in M} \hat{V}_{\texttt{left},i}^{(l)}, \arg\max_{j \in M} \hat{V}_{\texttt{right},j}^{(l)}$ for $\texttt{left}$ and $\texttt{right}$ operands respectively. Then, we perform operations $A \cup^* B$, $A \cap^* B$, $A -^* B$ and $B -^* A$. When $\forall_{l \leq L}\tau^{(l)} \approx 0$, both $\hat{\mathbf{V}}_{\texttt{left}}^{(l)}, \hat{\mathbf{V}}_{\texttt{right}}^{(l)}$ are one-hot vectors, and operations performed on occupancy values, as in Figure 2, are equivalent to CSG operations executed on aforementioned meshes, ex. by merging binary space partitioning trees of meshes [12]. Additionally, the whole CSG tree can be pruned to form binary tree, by investigating which meshes were selected through $\hat{\mathbf{V}}_{\texttt{left}}^{(l)}, \hat{\mathbf{V}}_{\texttt{right}}^{(l)}$ for the reconstruction, thus leaving the tree with $2^{L-l}$ nodes at each layer $l \leq L$.[2]

## 2.2 Training

The pipeline is optimized end-to-end using a backpropagation algorithm in a two-stage process.

**First stage** The goal is to find compositions of primitives that minimize the reconstruction error. We employ mean squared error of predicted occupancy values $\hat{\mathcal{O}}^{(L)}$ with the ground truth $\mathcal{O}^*$. Values are calculated for $\mathbf{X}$ which combines points sampled from the surface of the ground truth, and randomly sampled inside a unit cube (or square for 2D case):

$$\mathcal{L}_{\texttt{MSE}} = \mathbb{E}_{\mathbf{x} \in \mathbf{X}}[(\mathcal{O}^{(L)} - \mathcal{O}^*)^2] \tag{8}$$

We also ensure that the network predicts only positive values of parameters of shapes since only for such these shapes have analytical descriptions:

$$\mathcal{L}_{\texttt{P}} = \sum_{i=1}^{M} \sum_{p_i \in \mathbf{p}_i} \max(-p_i, 0) \tag{9}$$

To stop primitives from drifting away from the center of considered space in the early stages of the training, we minimize the clipped squared norm of the translation vector. At the same time, we allow primitives to be freely translated inside the space of interest:

$$\mathcal{L}_{\texttt{T}} = \sum_{i=1}^{M} \max(||\mathbf{t}_i||^2, 0.5) \tag{10}$$

The last component includes minimizing $|\alpha|$ to perform continuous binarization of distances into $\{\texttt{inside}, \texttt{outside}\}$ indicator values. Our goal is to find optimal parameters of our model by minimizing the total loss:

$$\mathcal{L}_{\texttt{total}} = \mathcal{L}_{\texttt{MSE}} + \mathcal{L}_{\texttt{P}} + \lambda_{\texttt{T}}\mathcal{L}_{\texttt{T}} + \lambda_\alpha |\alpha| \tag{11}$$

where we set $\lambda_{\texttt{T}} = \lambda_\alpha = 0.1$.

**Second stage** We strive for interpretable CSG relations. To achieve this, we output occupancy values, obtained with Equation 1, so these values create binary-valued sets since the $\alpha$ at this stage is near 0. The stage is triggered, when $\alpha \leq 0.05$. Its main goal is to enforce $\hat{\mathbf{V}}^{(l)}$ for $l \leq L$ to resemble one-hot mask by decreasing the temperature $\tau^{(l)}$ in CSG layers. The optimized loss is defined as:

$$\mathcal{L}_{\texttt{total}}^* = \mathcal{L}_{\texttt{total}} + \lambda_\tau \sum_{l=1}^{L} |\tau^{(l)}| \tag{12}$$

where we set $\lambda_\tau = 0.1$ for all experiments. Once $\alpha \approx 0$ and $\forall_{l \leq L}\tau^{(l)} \approx 0$, predictions of the CSG layers become fully interpretable as described above, i.e. CSG parse trees of reconstructions can be retrieved and processed using explicit representation of meshes. We also ensure that $\alpha$ and $\tau^{(l)}$ stay positive by manual clipping values to small positive number $\epsilon \approx 10^{-5}$, if they become negative. During experiments, we initialize them to $\alpha = 1$ and $\tau^{(l)} = 2$. Additional implementation details are provided in supplementary material.

# 3   Related Works

Problem of the 3D reconstruction gained momentum when the ShapeNet dataset was published [13]. The dataset contains sets of simple, textures meshes, split into multiple, unbalanced categories. Since then, many methods were invented for a discriminative [14–17] and generative applications [5, 6, 18, 19]. Currently, presenting results on this dataset allows the potential reader to quickly grasp how a particular method performs. There exists also a high volume ABC dataset [20] which consists of many complex CAD shapes. However, it is not well established as a benchmark in the community.

**3D surface representation**   Surface representations fall mainly into two categories: explicit (meshes) and implicit (ex. point clouds, voxels, signed distance fields). Many approaches working on meshes assume genus 0 as an initial shape that was refined to retrieve the final shape [21–23, 4, 24, 25]. Recent methods use step-by-step prediction of each vertex which position is conditioned on all previous vertices [26] and reinforcement learning to imitate real 3D graphics designer [27]. In Mesh-RCNN [28] a voxelized shape is retrieved first and then converted into mesh with the Pixel2Mesh [4] framework.

Implicit representations need an external method to convert an object to a mesh. 3D-R2N2 [1] and Pix2Vox [29] predict voxelized objects and leverage multiple views of the same object. These methods struggle with the cubic complexity of predictions. To overcome the problem, octree-based convolutional networks [30, 31] use encoded voxel volume to take an advantage of the sparsity of the representation.

Point clouds does not include vertex connectivity information. Therefore, ball-pivoting or Poisson surface reconstruction methods has to be employed to reconstruct the mesh [32, 33]. The representation is convenient to be processed using PointNet [14] framework. Objects can be generated using flow-based generative networks [34, 19].

Signed distance fields allow to model shapes with an arbitrary level details in theory. DeepSDF [3] and DualSDF [35] use a variational autodecoder approach to generate shapes. OccNet [36] and IM-NET [18] predict whether a point lies inside or outside of the shape. Such a representation is explored in BSP-NET [5] and CVXNET [6] which decompose shapes into union of convexes. Each convex is created by intersecting binary space partitions. Complexity of these methods provide high reconstruction accuracy but suffer from low interpretability in CAD applications. Convexes used in both methods are also problematic to modify from the perspective of a 3D graphic designer. Moreover, their CSG structure is fixed by definition. They use an intersection of hyperplanes first, and then perform union of predicted convexes.

Other approaches such as Visual Primitives (VP) [37] and Superquadrics (SQ) [38] base on a learnable union of defined primitives and provide high interpretability of results. However, superquadrics as primitives contain parameters that control shape and need to be on closed domain. Otherwise, distance function is not well-defined for them and learning these parameters become unstable.

**Constructive Solid Geometry**   CSG allows to combine shape primitives with boolean operators to obtain complex shapes. Much research is focused on probabilistic methods that find the most probable explanation of the shape through the process of inverse CSG [39] that outputs a parse tree. Approaches such as CSG-NET [7, 8] and DeepPrimitive [40] integrate finding CSG parse trees with neural networks. However, they heavily rely on a supervision. At each step of the parse tree, a neural network is given a primitive to output and a relation between primitives. The CSG-NET outputs a program with a defined grammar that can be used for rendering.

# 4   Experiments

We evaluate our approach on 2D autoencoding and 3D autoencoding tasks, and compare the results with state-of-the-art reference approaches for object reconstruction: CSG-NET [8] for the 2D task, and VP [37], SQ [37], BAE [41] and BSP-NET [5] for 3D tasks.

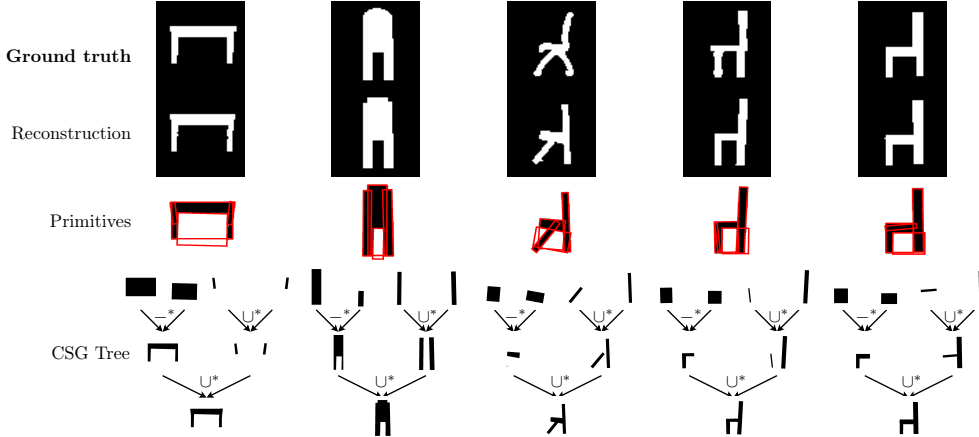

Figure 4: Reconstructions of the UCSG-NET from the CAD dataset. *Primitives* represent all these primitives that were used during CSG prediction, while *CSG Tree* - the parse tree of the reconstruction.

## 4.1 2D Reconstruction

For this experiment, we used CAD dataset [7] consisting of 8,000 CAD shapes in three categories: chair, desk, and lamps. Each shape was rendered to $64 \times 64$ image. We compare our method with the CSG-NETSTACK [8], improved version of the CSG-NET [7], on the same validation split. Table 1 contains comparison with CSG-NET working in both modes. Following the methodology introduced in existing reference works, methods are evaluated on Chamfer Distance (CD) of reconstructions. We set 2 CSG layers for our method, where each outputs 16 shapes in total. The decoder predicts parameters of 16 circles and 16 rectangles. Our method, while being fully unsupervised, is better

Table 1: **Reconstruction performance on CAD dataset** – We evaluate our method and compare it with CSG-NETSTACK [8]. $k$ denotes the beam search size and $i$ the number of refinement steps of the reconstruction. $i = \infty$ signifies the refinement until the reconstruction quality converges. Our method works naturally with $k = 1$ and $i = 0$.

| Method | Mode | $k$ | $i = 0$ | $i = \infty$ |
|---|---|---|---|---|
| CSG-NETSTACK | Supervised | 1 | 3.98 | 2.25 |
| CSG-NETSTACK | Supervised | 10 | 1.38 | 0.39 |
| CSG-NETSTACK | RL | 1 | 1.27 | 0.57 |
| CSG-NETSTACK | RL | 10 | 1.02 | **0.34** |
| Our | Unsupervised | 1 | **0.32** | - |

then the best variants of CSG-NET and is significantly better with no output refinement. Results show that the method is able to discover good CSG parse trees without explicit ground truth for each level of the tree. Therefore, it can be used where such ground truth is not available.

We present qualitative evaluation results in Figure 4 and visualize used shapes for the reconstruction. The UCSG-NET uses proper operations at each level that lead to the correct shape reconstruction. In most cases, it puts rectangles only. The nature of the dataset causes that phenomenon. To avoid possible errors, the network often uses a union of overlapping shapes to pass the primitive untouched.

## 4.2 3D Autoencoding

For the 3D autoencoding task, we train the model on $64^3$ volumes of voxelized shapes in the ShapeNet dataset. We sample 16384 points as a ground truth with a higher probability of sampling near the surface. To speed up the training, we applied early stopping heuristic and stop after 40 epochs of no improvement on the $\mathcal{L}^*_{\texttt{total}}$ loss. The data was provided by Chen et al. [5] and bases on the 13 most common classes in the ShapeNet dataset [13]. We used 5 CSG layers to increase the diversity

Table 2: **3D reconstruction quality** measured as Chamfer Distance on 3D autoencoding task.

|  | High interpretability | | Low interpretability | |
| --- | --- | --- | --- | --- |
|  | **Ours** | VP [37] | SQ [38] | BAE [41] | BSP-Net [5] |
| Chamfer Distance | 2.085 | 2.259 | 1.656 | 1.592 | **0.446** |

of predictions and set 64 parameters of spheres and boxes to handle the complex nature of the dataset. Each layer predicts CSG 48 combinations of these primitives. Training takes about two days on Nvidia Titan RTX GPU. The CSG inference for a single sample takes 0.068s and the reconstruction - 1.68s using the `libigl` library.

We follow the procedure described in [5] and report Chamfer Distance as a quality measure of the reconstruction. We evaluate it on 4096 points sampled from the surface of the reconstructed object. We reconstruct shapes from CSG trees retrieved from predictions of our model. Obtained results are shown in Table 2. Examples of reconstructed shapes are presented in Figure 5. We can see that it accurately reconstructs the main components of a shape which resembles Visual Primitives (VP) [37] approach where outputs can be treated as shape abstractions. The remaining reference approaches

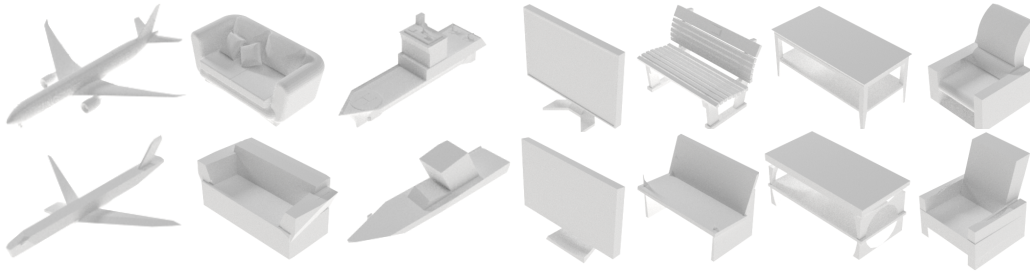

Figure 5: **Surface reconstructions** of UCSG-NET on 3D autoencoding tasks. Top row: ground truth. Bottom row: reconstructions.

outperformed our model with respect to CD measure. It was mainly caused by failed reconstructions of details, such as engines on wings of airplanes, to which the metric is sensitive. However, our ultimate goal was to provide an effective and interpretable method to construct a CSG tree with limited number of primitives.

Finally, we show an example parse tree in Figure 6, used to reconstruct an example shape from the validation set. The model manages to create diverse combinations of primitives and reuse them at any level. Since many primitives were used in later layers, the tree complexity is not necessarily $2^L$. Notice that the main body and wings were reconstructed separately. We found that the model learns to reconstruct particular semantic parts of the object separately, for example, wings and the hull of an airplane or legs and the counter of a desk. These parts are merged in the final CSG layer where we force a union operation to be performed. See the supplementary material for additional CSG tree visualizations.

## 5 Conclusions

We demonstrate UCSG-NET - an unsupervised method for discovering constructive solid geometry parse trees that composes primitives to reconstruct an input shape. Our method predicts CSG trees and is able to use different Boolean operations while maintaining reasonable accuracy of reconstructions. Inferred CSG trees are used to form meshes directly, without the need to use explicit reconstruction methods for implicit representations. We show that these trees can be easily visualized, thus providing interpretability about reconstructions step-by-step. Therefore, the method can be applied in CAD applications for quick prototyping of 3D objects.

We identified three interesting venues to be taken in future works. In one of them, we would incorporate weak supervision to provide hints to the network what CSG operations are expected to be used for a particular shape. Since there are many CSG trees that reconstruct the same object and

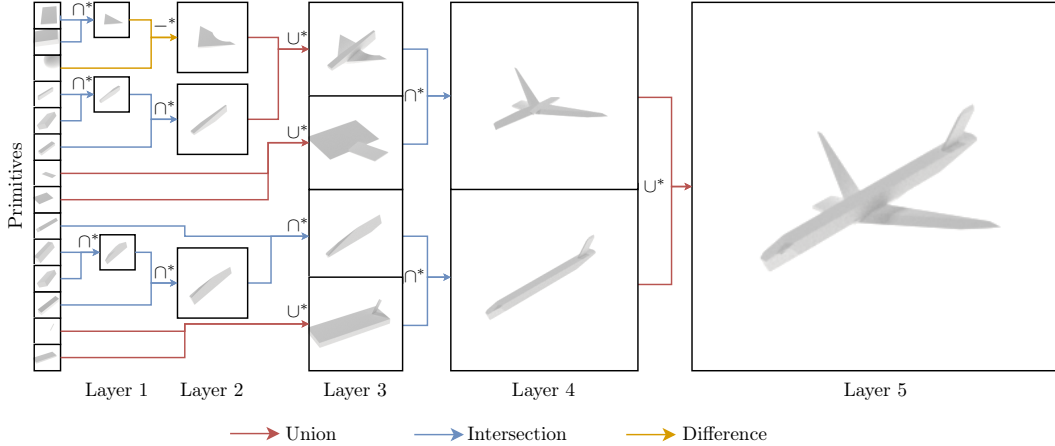

Figure 6: **Retrieved CSG tree** from the reconstruction prediction of an airplane from the validation set. We noticed that the model approximates all airplanes with a similar CSG tree.

the space of solution is vast, such a supervision can improve the final results. Other paths include: using efficient RANSAC [42] to provide initial primitives, formulating a single CSG layer as a Set Transformer [43] or applying regularization techniques known in transformers [44] to increase diversity of predicted CSG trees.

## 6 Acknowledgments

We thank the reviewers for their insightful comments that led us to improve the final manuscript. This work was supported in part by the National Science Centre, Poland research project no. 2016/21/D/ST6/02948, statutory funds of Department of Computational Intelligence and by Microsoft Research. We also acknowledge the support of NVIDIA Corporation with the donation of the Titan Xp GPU used in a part of the research.

## Broader Impact

UCSG-NET can find applications in CAD software. When applied, it is possible to retrieve a CSG parse tree for a particular object of interest. Hence, for a situation when a 3D object was modeled with a sculpting tool, the model can approximate it with single primitives and operations between them. Then, such a reconstruction can be integrated into existing CAD models. We find that beneficial in speeding up the prototyping process in 3D modeling.

However, inexperienced CAD software users can rely heavily on presented assumptions. In the era of 3D printing ubiquity, printed elements out of reconstructed CSG parse trees can be erroneous, thus breaking the whole item. Therefore, we note that integrating our method into existing software should serve mainly as a prototyping device.

We encourage further research on an unsupervised CSG parse tree recovery. We suspect that this area stagnated due to constraining limitations that a CSG tree creates a single object, but a single object can be created out of infinity many CSG trees. Therefore, new methods need to be invented that provide good approximations of CSG trees with short inference times.

## Footnotes

*Now at Warsaw University of Technology

[1]We published our code at https://github.com/kacperkan/ucsgnet

[2]We consider the worst case, since some shapes can be reused in consecutive layers, hence number of used shapes in the layer $l$ can be less than $2^{L-l}$.

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
