[Supplementary Material]

# A Mathematical Formulation of Primitives in the Signed Distance Field Representation

All used primitives are represented as signed distance fields $\mathcal{D}$. It means, that instead of having a discretized mesh, we evaluate distance of any point $\mathbf{x}$ to the surface of the object. Such a formulation provides continuous representation of an object. Affine transformation of SDF objects means performing an inverse of the same affine transformation on a point $\mathbf{x}$ and then evaluating the distance.

There exist plethora of primitives that can be defined in SDF, however we focus mainly on using squares and circles for the 2D data or boxes and spheres for the 3D data. In Table 3, we show mathematical formulations of these primitives.[3]

Table 3: Mathematical formulations of primitives in the SDF representation for 2D (left) and 3D (right). We assume that each shape is described by parameters $\mathbf{p}$ and we evaluate it for a point $\mathbf{x}$. $|\cdot|$ denotes absolute value and $||\cdot||$ - $L_2$ norm. All shapes assume center of mass at the point $\mathbf{0}$.

| Shape | Formula | Shape | Formula |
|---|---|---|---|
| Rectangle | $\mathbf{p} = [p_x, p_y]$ <br> $\mathbf{q} = |\mathbf{x}| - \mathbf{p}$ <br> $\mathcal{D} = \|\max(\mathbf{q}, 0)\|$ <br> $\quad + \min(\max(q_x, q_y), 0)$ | Box | $\mathbf{p} = [p_x, p_y, p_z]$ <br> $\mathbf{q} = |\mathbf{x}| - \mathbf{p}$ <br> $\mathcal{D} = \|\max(\mathbf{q}, 0)\|$ <br> $\quad + \min(\max(q_x, \max(q_y, q_z)), 0)$ |
| Circle | $\mathbf{p} = [p_r]$ <br> $\mathcal{D} = \|\mathbf{x}\| - p_r$ | Sphere | $\mathbf{p} = [p_r]$ <br> $\mathcal{D} = \|\mathbf{x}\| - p_r$ |

# B Implementation details

We follow architectures described by Chen et al. [5] to show influence of our framework on obtained results. For 2D and 3D autoencoding, we use a simple convolutional network where each convolutional layer reduces feature map spatial dimensions by a factor of two. The decoder is a multilayer perceptron with a leaky ReLU activation units in each hidden layer. The final layer outputs parameters of primitives and its size varies depending on a number of considered dimensions, a number of input and output shapes. Refer to Section 2 for more details. No batch normalization is used. The parameter prediction network takes the latent code of size $d_{\mathbf{z}} = 256$. Parameter encoders $h^{(l)}$ consists of a single hidden fully connected layer of size $d_{\mathbf{z}}$. GRU units has a latent dimension size equal to $d_{\mathbf{z}}$. Architectures are summarized in Table 4. In CSG layers we sample initial values of $\mathbf{K}_{\text{left}}^{(l)}, \mathbf{K}_{\text{right}}^{(l)}$ from $\mathcal{N}(0, 0.1)$. We use Adam optimizer [45] for each task with learning rate $10^{-4}$ and beta parameters $(0.5, 0.99)$. Batch size was set to 16 samples in 2D and 3D autoencoding tasks. Learning starts with initial values $\alpha = 1$ and $\tau^{(l)} = 2$. We use 2 CSG layers for the 2D data and 5 for the 3D.

Table 4: Architectures used during experiments of the encoder (left), decoder (middle) and $h^{(l)}$ (right). `fc` stands for a fully connected layer, `conv` - a convolutional layer, `flatten` - a layer that merges all dimensions of the input tensor into a single vector. Each convolutional layer has kernel of size $4 \times 4$, stride 2 and leaky ReLU activation with a slope 0.01. All layers include bias terms.

| Layer | Out features | Padding | Layer | Out features | Layer | Out features |
|---|---|---|---|---|---|---|
| conv1 | 32 | 1 | fc1 | 512 | fc1 | 256 |
| conv2 | 64 | 1 | fc2 | 1024 | fc2 | 256 |
| conv3 | 128 | 1 | fc3 | 2048 | | |
| conv4 | 256 | 1 | | | | |
| conv5 | 256 | 0 | | | | |
| flatten | 256 | - | | | | |

# C Constructive Solid Geometry Tree Visualizations

Figure 7: Visualization of CSG trees of reconstructed objects. → denotes the union operation, → - the intersection and → - the difference. Our model is able to reconstruct in the same manner objects that are semantically similar, such as the legs of a chair and table.

| Layer 1 | Layer 2 | Layer 3 | Layer 4 | Layer 5 |

Figure 8: Visualization of CSG trees obtained with a new set of weights trained from scratch. Note that at each new training, the model learns CSG structures.

# D Diversity of CSG structures - Discussion

UCSG-NET tends to learn a single CSG tree structure for different samples in the dataset. Conditioning of matrices $\mathbf{K}^{(l)}_{\text{left}}$ and $\mathbf{K}^{(l)}_{\text{right}}$ on $\mathbf{z}^{(l)}$, applying common regularization practices such as dropout or $L_2$ norm led to worse quantitative results, hence we did not apply them in the final version of the model.

When the model was trained, we found the following trend. In the first stage, it focused on learning parameters of primitives to minimize the reconstruction error. At that point, CSG structures were diverse but some layers tended to select multiple shapes with equal probability. When the second stage was triggered, the network converged to a single structure.

To support our claim that the network can learn different trees in a single run, we show CSG structures in Figure 9 from the middle of training on the CAD dataset. We leave enforcing diverse structures across samples for future work.

Figure 9: **Predicted CSG trees** from a different training run of UCSG-NET for 2D CAD dataset.

## Footnotes

[3]More primitive formulations can be found at `https://www.iquilezles.org/www/articles/distfunctions/distfunctions.htm`