[Reviews · NeurIPS 2020]

Review 1

Summary and Contributions: This work proposes a interpretable CSG parsing network without explicit CSG parse tree as the supervision. Existing approaches along this direction are supervised and require the entire CSG parse tree that is given upfront during the training process. This work is evaluated on 2D and 3D auto-encoder tasks, and it shows reasonable results.

Strengths: I believe the idea is novel and very interesting. It can train the network to generate CSG parse tree without explicit CSG parsing tree as the supervision. This work is applicable to a broader set of datasets (and thus applications) comparing to CSG-NET.

Weaknesses: The reconstruction is lack of details, which leads to inferior results in the 3D task comparing to the SOTA. I think further learning a SDF for the local primitives might help to solve this problem. The major advantage of this work over BSP-Net (and other SOTA) is its interpretability, and thus I think it should add more discussion about the interpretability. Since BSP-Net generate a BSP tree, comparing the generated tree might help the authors to understand the difference and show the value of this work.

Correctness: Mostly correct

Clarity: Well written

Relation to Prior Work: Good

Reproducibility: Yes

Additional Feedback: After reading the other reviewers' comments and rebuttal, I would like to keep my original score. This work proposes an interesting idea and has good techinque contributions, and I think it is well motivated. Though I also agree that it does not show impressive results comparing to SOTA.


Review 2

Summary and Contributions: Given an input 3D shape (or 2D mask), this paper aims to predict a CSG representation that explains the input. The proposed approach is to a) encode the input and generate a set of primitives at the initial layer, and b) hierarchically build a CSG tree via proposed CSG layers, and c) compose the entities output via the last layer to obtain the predicted shape. The key contribution here is the idea of the CSG layer which, given some entities and a shape embedding as input, produces a set of output entities by computing union/intersection/difference of (multiple) pairs of input entities, where the decision of which pairs to select adaptively depends on the shape encoding. In sense, I think this approach is a generalization of BSPNet where combinations beyond intersections are allowed. The overall approach allows learning CSG predictions in an unsupervised manner and results are presented across different categories.

Strengths: - I really like the overall idea. The key insight of having a CSG layer that can yield different shapes via operating on selected pairs is very simple, elegant and novel. I essentially view this as a generalization of BSPNet which only considered half-space intersections, whereas this paper generalizes to more operations (and a broader set of base primitives) to allow predicted CSG tree. - In context of predicting a CSG representation, I think this is the first framework I've seen that allows learning without supervision. The previous CSG prediction approaches e.g. CSGNet relied on having the full shape programs that generated the shapes whereas this work obviates the need of such supervision.

Weaknesses: - While I theoretically like the approach, the empirical results are unfortunately not very convincing. In particular, the approach seems to indicate (in L102-104) that different pairs can be selected based on the input embedding Z, this is not apparent from the examples. As an illustration, all the results shown in Figure 3 have the exact same CSG tree. I am therefore not convinced that this approach does indeed predict different CSG trees for different instances. - On a point related to above, for 3D shape inference, only a single example is shown per category (including supplementary and main text) and this is not convincing. Ideally, the paper should show several random examples per category of the obtained CSGs to convince the reader e.g. how are chairs with 4 legs vs chairs with wheels handled. - The proposed representation seems to be empirically worse at representing the shapes than the arguably simpler approach of composing superquadrics and therefore the experiments are not convincing of why one would leverage this CSG representation. I feel that this maybe partly due to the data considered - ShapeNet objects are generally union of primitives, whereas this paper's strength is in also allowing operations like difference. Towards highlighting this better, I would strongly recommend considering alternate data to showcase this method e.g. 'ABC Dataset'.

Correctness: The technical details and empirical setup seem correct.

Clarity: The paper is generally well written and easy to follow.

Relation to Prior Work: I think the relation to prior work is presented accurately.

Reproducibility: Yes

Additional Feedback: This paper presents an interesting idea from a technical perspective but the empirical results are unfortunately not convincing as: a) It is not clear if CSG tree prediction really works or whether just a single CSG tree template has been learned per category, and b) the proposed approach is worse at representing shapes than arguably simpler methods e.g. SQs and I think this approach needs to be evaluated on datasets where CSGs are more directly required. Overall, I think this is a promising direction but feel the paper is not quite ready yet from an empirical perspective. ---- Update after response: Thank you for the response. I agree with the other reviewers that the results don't need to be better than (or even close to) SOTA for this paper's acceptance. That said, my main concern was not regarding whether the approach works better than prior work, but rather whether it works even when viewed on its own i.e. can it learn prediction of meaningful CSG trees across different instances? All the results shown in the main text did depicted the same CSG tree across instances, and therefore it was not clear that the approach 'works' as one would intuitively like it to. The author response addresses this to some extent by showing a couple of examples in the 2D mask setting where different CSG trees are predicted, but I would have ideally liked evidence in slightly more challenging settings (e.g. 3D chairs) and without handpicking results (random samples). Overall, I also really like the formulation and think it is interesting and possible for future efforts to build upon. The main question is where does one draw the line in terms of 'sufficient empirical evidence' and unfortunately for me the paper is still below that rather subjective line. Overall, while I'd personally not recommend acceptance, I think the paper is technically interesting and novel enough that I'd not argue against it as well.


Review 3

Summary and Contributions: This paper proposes the first neural method to create a CSG tree to represent an input shape without supervision. CSG representation is highly appealing, since it is more interpretable than most of the other neural representations and can be used as a starting point for interactive modeling. The downside, is that the reconstruction accuracy seems quite low.

Strengths: Training CGS prediction without explicit supervision is an important contribution. In 2D case, the method seems to outperform the strongly-supervised variant (CSG-Net).

Weaknesses: This paper should include a more thorough discussion on the technical differences from CSG-Net. The high-level contribution (i.e., ability to train in an unsupervised fashion) is clear, but what's not clear is what are the low-level representation decisions that enabled this.

Correctness: Yes.

Clarity: I found the paper to be somewhat difficult to follow. Figure 3 seems to provide a nice illustration, but it's hardly referenced in the text. I also wish the paper explicitly articulated the low-level technical differences from CSG-Net.

Relation to Prior Work: Mostly yes, I agree that even if reconstruction quality is lower than some of the recent techniques (e.g., BSP-Net), there is an inherent value to using CSG as high-level and interpretable representation. There is not much competition in that space (and it looks like it's difficult to achieve high-quality results via existing methods, including this submission), but it's a promising direction. The big glaring omission is the discussion on the relationship between UCSG-Net and CGS-Net [7]. Basically, at a high level the former is unsupervised and the latter is supervised, but it's not clear what are the critical technical differences.

Reproducibility: Yes

Additional Feedback:


Review 4

Summary and Contributions: This paper proposes a method to, in an unsupervised manner, learn a constructive solid geometry parse tree that represents 2D/3D shapes. The model first predicts a set of implicit primitives from an encoding of the input shape. It then learn to apply CSG operations in a bottom-up, hierarchical way to create the final output shape. The model is trained with interpretability in mind, and the results show that the model can achieve good performances on 2D/3D reconstruction tasks and the resulting CSG tree is of acceptable quality, though not as good as fully supervised ones.

Strengths: -A novel enough idea that combines the strengths of two lines of work: primitive-based shape decomposition and (supervised) CSG tree parsing. The resulting paradigm decomposes shapes into more interpretable primitives than works such as BSP-NET, whereas also require much less supervision than works such as CSG-NET. -Sound technical details that all seem important to the ultimate goal of the method (accurate and interpretable CSG trees). -Good evaluation protocols that compares the method accurately with relevant prior works.

Weaknesses: -The results are not clearly superior, both quantitatively (worse reconstruction than some prior works) and qualitatively (some of the results, especially 3D, look questionable). It is also worth noting that the resulting CSG-trees are often redundant and qualitatively dissimilar to human created ones. However, I feel that all these limitations are acceptable for a paper that aims to establish a momentum in a new direction. -The evaluation mostly focuses on reconstruction quality, but since the goal is interpretable CSG programs, more evaluations on interpretability / similarity to human made programs are desirable.

Correctness: I believe the technical details are correct but am not completely sure. The empirical comparisons are standard and thorough.

Clarity: The paper is in general well written and easy to follow.

Relation to Prior Work: The authors mentioned the relevant prior works alone the two line of researches this work aim to combine. The differences from each line of work is stated clearly, with sound motivation on the direction of improvement.

Reproducibility: Yes

Additional Feedback: I tend to accept this paper because it demonstrates a very promising probability - that it is now possible to learn a relatively complex shape representation (CSG) that are often used in actual production settings. Granted, the novelty is slightly limited (the training protocol is similar to earlier works in implicit shape generation/reconstruction and the representation (CSG) is also widely used), and the results are not super convincing (will explain below); however, I still feel the idea is interesting enough and will easily sparkle future works in similar directions. I thus lean on the acceptance side. I am not completely positive because I am uncertain if the method leads to a dead end - as there are infinitely many possible CSG trees for a given shape, an unsupervised method might never be able to learn something that is truly usable, even after many improvements over the current method. Additional comments: -Could the authors explain the motivation of adopting a bottom-up process that groups primitive to form final outputs? It seems to me that it would be much harder for the network to understand shape decomposition in this way, since it is trying to select form a large set of primitives and find which ones can be grouped to form something that look similar to the input, rather than directly thinking about how to decompose the input shape (in a top-down way). This is a lesser problem for method such as BSP-NET that does not focus on interpretability, but since the current works aims at creating interpretable CSG programs, I feel that this might result in limitations that are hard to address. - As mentioned in earlier sections, reconstruction loss is not the only thing that matters for CSG-trees. Judging from the qualitative examples, I think many trees are suboptimal and the way the networks creates certain shapes are quite problematic (e.g. Figure 4, 4th example, Figure 6). Comparing the generated CSG trees to human created ones will be helpful here, even if the results are expected to be worse due to the lack of direct supervision. -Speaking of supervision, (just curious), have the authors thought about ways to add some weak supervision to the networks to let it know what CSG operations are desirable and what are bad (for example, I think people rarely ever use intersection in production...) -L251 "...contribute to learning disentangled representation of parts..." I am not convinced by a few qualitative examples that this claim holds. Can the authors provide some summary statistics showing that different semantic parts are often reconstructed separately? ====Post-rebuttal comment===== I thank the authors for answering my (mostly minor) concerns. While I agree with other reviewers that the quality of the results are not outstanding, I hold my opinion that this work provides momentum towards a novel direction and would keep my rating of acceptance.

[Author Response · NeurIPS 2020]

We thank the reviewers for their insightful comments. In this rebuttal, we respond to remarks from reviews. If accepted, we will extend the submission with discussions from below.

**REVIEWER #1**

**Remark 1** The work lacks discussion about the comparison of interpretability with BSP-Net.

**Response** BSP-Net generates multiple convex parts that are difficult to interpret due to an unbounded number of possible vertices to be used by each convex. These convexes are also problematic to modify from the perspective of a 3D designer. Moreover, their CSG structure is fixed by definition. It uses the intersection of hyperplanes first, and then the union of convexes.

**REVIEWER #2**

**Remark 1** While the UCSG-NET can predict different CSG trees for different object instances, an empirical evaluation does not show that.

**Response** Our approach is capable of generating diverse CSG trees for different instances (see Figure on the right). We found empirically, that the diversity of CSG trees can be increased with a layer normalization applied for $\{l \in L | \mathbf{z}^{(l)}, h^{(l)}(\hat{\mathbf{V}}^{(l)})\}$. However, this operation slightly degrades quantitative results and we omitted it in our submission. We sacrificed the diversity of CSG trees for accuracy and reported the qualitative and quantitative results for the best model.

**Remark 2** Only a single instance of CSG visualization for each class is shown.

**Response** We will provide more various CSG visualizations for each class in the supplementary (including Remark 1).

**Remark 3** The proposed approach is inferior to the seemingly more straightforward superquadrics method. The structure of the ShapeNet dataset may cause it, and further experiments on the ABC dataset would be needed.

**Response** Superquadrics [36] method follows the evaluation methodology of Visual Primitives (VP) [37]. These two approaches use only a union operation. While our approach is inferior to the superquadrics, we focused on the applicability of our method in 3D design processes where the CSG is commonly used. Therefore, we argue that referenced approaches are not as versatile as ours. Each work was evaluated on ShapeNet as it is a standard benchmark. Presenting results on this dataset allows the reader to quickly grasp how the method performs in terms of the reconstruction. Referring to ABC, we will put effort into analyzing the dataset to use it in our approach.

**REVIEWER #3**

**Remark 1** It is unclear, what low-level representation decisions allowed UCSG-NET, in contrast to CSG-NET, to be trained in an unsupervised fashion.

**Response** The CSG-NET predicts a 3D program. Since the program's instructions are discrete, the model is unable to be smoothly optimized towards a particular set of instructions. Therefore, the direct training in an unsupervised manner would be difficult. To solve this limitation, we proposed the following advancements over CSG-NET. Firstly, we applied smooth boolean operations that are pushed towards discrete forms during the training. Secondly, we used the SDF representation of shapes. It allowed us to convert shapes into occupancy values and apply differentiable CSG operations (Fig. 2). These operations are selected dynamically through the attention mechanism (Fig. 3.). Hence, UCSG-NET does not need any supervision to guide which CSG operations should be selected.

**REVIEWER #4**

**Remark 1** Predicted CSG trees are often redundant and qualitatively dissimilar to human-created ones.

**Response** The network predicts such shapes that would not worsen the final reconstruction. Hence, it can predict highly overlapping boxes. At the same time, we do not force any particular structure of the tree to be learned. Since the task is ill-posed and there exist infinite CSG trees that reconstruct the same shape, we argue that the space solutions can be constrained by weak supervision that would enhance the fidelity of predicted CSG trees.

**Remark 2** Could the authors explain the motivation of a bottom-up process that groups primitive to form final outputs?

**Response** To our knowledge, there are not many machine learning approaches that apply top-down decomposition of meshes that have proven high reconstruction quality. Moreover, our method was designed in such a way that it can be used as an extension for other bottom-up approaches in the literature. Therefore, we believe that our work will encourage future research further to investigate applicability of introduced CSG layers.

**Remark 3** Can weak supervision be used in the introduced method?

**Response** Applying weak supervision is an interesting future direction. Possibly, it would require introducing a new term in $\mathcal{L}_{\texttt{total}}$. We would also limit the number of possible primitives to be predicted, match primitive predictions with the ground truth using the Hungarian method, and use cross-entropy loss to optimize predicted masks $\hat{\mathbf{V}}^{(l)}$ directly.

**Remark 4** The sentence "contribute to learning disentangled representation of parts" from seems to be unsupported.

**Response** To clarify, we found empirically, that our method often separates particular semantic elements of the object during reconstruction (hence disentanglement) and merges them in the final layer, ex. wings and the hull of an airplane or legs and the counter of a desk.

[Meta-Review · NeurIPS 2020]

All reviewers are enthusiastic about the idea of the paper. One reviewer is concerned about sufficient empirical validation, but states that it is a matter of where is the threshold for sufficient empirical validation. AC finds the paper sufficiently novel and interesting, and agrees with the reviewers that it points to a promising future direction. Thus AC recommends acceptance. In the camera ready version, in addition to revising the paper based on the rebuttal, the authors should pay particular attention to strengthening empirical validation. In particular, the authors should include more results to show stats on the types of trees that it learns and provide a better description on how well the method works.